



# Quantifying primary oxidation products in the OH-initiated reaction of benzyl alcohol

Reina S. Buenconsejo[1], Sophia M. Charan[1], John H. Seinfeld[1], and Paul O. Wennberg[1]

[1]California Institute of Technology, Pasadena, CA 91125

**Correspondence:** Paul O. Wennberg (wennberg@caltech.edu)

**Abstract.** Benzyl alcohol is a compound found in many volatile chemical products (VCPs), and is widely used in personal care products and as an industrial solvent. While past work has empirically identified oxidation products, we do not understand explicit branching ratios for first-generation benzyl alcohol oxidation products, particularly over a range of [NO] conditions. Using gas chromatography (GC) in tandem with chemical ionization mass spectrometry (CIMS), we measure the branching

ratios of major oxidation products, namely hydroxybenzyl alcohol (HBA) and benzaldehyde. Later-generation oxidation products from both HBA and benzaldehyde pathways are also observed. We find the H-abstraction route leading to benzaldehyde formation unaffected by [NO], with a branching ratio of ~ 19%. The OH addition route, however, which leads to HBA formation, does vary with [NO]. At higher [NO], we report a branching ratio for HBA of ~ 45 − 47% and as high as ~ 69% at low [NO]. We also find that HBA has a high secondary organic aerosol (SOA) yield and, therefore, likely contributes to the

high SOA yield of benzyl alcohol which under some conditions can approach unity. Insights from the present study can help elucidate the chemistry of other atmospherically-relevant aromatic compounds, especially those found in VCPs.

## 1 Introduction

Recent work indicates the growing importance of volatile chemical products (VCPs) in driving air pollution chemistry, particularly as regulations decline the contribution of vehicular-based emissions (McDonald et al., 2018; Coggon et al., 2021). VCPs

play an important role in air quality because of their high potential to form secondary organic aerosol (SOA); an analysis of VCPs in the Los Angeles Air Basin indicates that VCPs could contribute up to 70% of SOA formation in the Los Angeles Air Basin despite accounting for only approximately 4% of total petrochemical product use (McDonald et al., 2018). Regional scale modeling that includes VCP emission inventories also shows improved agreement with ambient data compared to past models that historically did not consider VCP emissions (Seltzer et al., 2021; Pennington et al., 2021). However, because many

of the chemicals comprising VCP emissions have not been studied in laboratory settings, their contribution to SOA formation is uncertain. Therefore, additional experimental work on VCPs is still needed.

Here, we evaluate the photochemistry of benzyl alcohol, a compound found prominently in VCPs. Benzyl alcohol is used in soaps and perfumes, and is also used as a solvent in the manufacture of paints, inks, lacquers, and epoxies (Wang, 2015). In a previous study, the aerosol mass yield of benzyl alcohol was found to have a high mass yield under a variety of conditions–even

approaching unity (Charan et al., 2020; Jaoui et al., 2023).



Experimental studies, including kinetic experiments on benzyl alcohol, have identified the primary atmospheric oxidative pathway of benzyl alcohol to proceed via reaction with the OH radical (Bernard et al., 2013; Harrison and Wells, 2009). Harrison et al. calculated that benzyl alcohol reacts with OH at a rate of $(28 \pm 7) \times 10^{-12}$ cm$^3$molecule$^{-1}$s$^{-1}$. The reaction rate with O$_3$ is too slow to contribute to its oxidation in most environments (Harrison and Wells, 2009).

Previous experimental studies of benzyl alcohol identified several oxidation products including hydroxybenzyl alcohol (HBA) and benzaldehyde (Burkholder et al., 2017; Harrison and Wells, 2009; Bernard et al., 2013). A theoretical study on the mechanism of benzyl alcohol oxidation initiated by the hydroxy radicals (OH) also predicted the major oxidation pathways to form benzaldehyde, i-hydroxybenzyl alcohol, and o-hydroxybenzyl alcohol (Wang, 2015). Other products observed in past experimental studies include ring-opening products, such as but-2-enedial and 6-hydroxy-5-oxohex-2-enal (Harrison

and Wells, 2009), as well as C6 compounds, such as dihydroxy benzene (Bernard et al., 2013). These past experiments were conducted under high NO conditions. Yet, understanding chemical mechanisms and yields as a function of NO is important for understanding urban air quality as NO$_x$ to VOC ratios have been shown to affect OH oxidation of VOCs and as NO$_x$ regimes continue to change in urban settings (Seinfeld and Pandis, 2016; Parker et al., 2020). This is particularly relevant as NO$_x$ continues to decline in many U.S. urban areas (Parker et al., 2020).

In this study, we draw on past work which suggests HBA forms via addition of OH to the aromatic ring. We experimentally confirm products identified in past work, including products predicted in a theoretical study (Wang, 2015). We also report quantitative values for HBA, benzaldehyde, phenol, and hydroxy oxopentenal. We find the branching ratios for HBA, the primary oxidation product, depend on [NO] and range from $\sim 45-69\%$. Benzaldehyde, the other main oxidation product, has a branching ratio of $\sim 19\%$.

## 45  2   Methods

Chamber experiments were conducted to elucidate the chemical mechanism of benzyl alcohol oxidation via OH. First-generation oxidation products were also used to identify important pathways to SOA formation. Two set-ups were used to investigate: 1.) Gas-phase yields and gas-phase oxidation chemistry; and 2.) The chemical pathways responsible for SOA formation.

### 2.1   Experimental Design

Gas-phase experiments were conducted in a $\sim 0.8$ m$^3$ FEP Teflon-walled environmental chamber, henceforth referred to as Chamber G. The chamber was filled and evacuated multiple times with purified air prior to experiments. Particle-phase experiments were conducted in a 19 m$^3$ FEP Teflon-walled environmental chamber (henceforth referred to as Chamber P) which was continuously flushed for $> 24$ h prior to experiments. All experiments were run at room temperature ($\sim 22°$C), low relative humidity ($< 10\%$ RH), and ambient pressure ($\sim 1$ atm). Benzyl alcohol (Sigma Millipore, ReagentPlus $\geq 99\%$) and benzaldehyde

(Sigma Millipore, ReagentPlus $\geq 99\%$) were injected into the chamber by flowing warm air over over a measured amount of precursor deposited on a Pall Teflon filter. For gas-phase experiments, 2-hydroxybenzyl alcohol (HBA) (Sigma Millipore $99\%$) was injected similarly. Because HBA is a relatively low-volatility compound, in order to achieve sufficiently high concentra-





tions in Chamber P for particle-phase experiments, HBA was dissolved in milliQ water and injected by bubbling purified air through the solution.

For high NO gas-phase experiments (G1 – G2), methyl nitrite was used as the oxidant precursor by measuring the pressure of methyl nitrite into an evacuated round-bottom bulb and back-filling the remainder of the bulb volume with nitrogen. NO (1993±20 ppm NO in $N_2$, Matheson) was prepared in a similar manner. For high NO, particle-phase experiments, NO (506.9±10 ppm NO in $N_2$) was injected using a mass-flow controller (Sierra Instruments).

In experiments G3 and P1 – P2, hydrogen peroxide ($H_2O_2$) (Sigma Millipore, 50 wt% in $H_2O$ stabilized) was used as the 65 OH precursor by injecting a known mass into a glass bulb and flowing purified air over the liquid droplets. In particle-phase experiments, the $H_2O_2$ was heated in a water bath (~ 42°C) during injection.

All particle-phase experiments were seeded using a sonicated solution of 0.06 M ammonium sulfate (($NH_4$)$_2SO_4$). During injection, the seed solution was run through a soft x-ray charge conditioner (TSI Model 2088). In all experiments, ~ 1 h was allowed after all injections were completed and before the lights were turned on to allow for mixing and collection of adequate 70 background data. In particle-phase experiments, this time was also used to confirm previously calculated chamber wall loss parameters (Charan et al., 2018). Ultraviolet (UV) broadband lights centered around ~ 350 nm were used as the light source for photooxidation (Light Sources, Inc.); oxidation duration was determined so that ~ 10% of the precursor was reacted in gas-phase reactions. The $j_{NO2}$ and $j_{CH3ONO}$ for Chamber G are $4.4 \times 10^{-3}$ s$^{-1}$ and $1.1 \times 10^{-3}$ s$^{-1}$, respectively. The $j_{NO2}$ and $j_{H2O2}$ of Chamber P are measured to be $4.4 \times 10^{-3}$ s$^{-1}$ and $3.2 \times 10^{-6}$ s$^{-1}$, respectively, using methods described elsewhere 75 (Zafonte et al., 1977). Additional information about the oxidants is found in Appendix A. Photooxidation was carried out for particle-phase experiments for > 6 hrs. This oxidation time corresponds to ~ 50 – 100% reaction of the initial VOC and is congruent with reaction times in the benzyl alcohol studies in (Charan et al., 2020). A summary of experiments can be found in Table 1.



**Table 1.** Experimental Summary.

| Expt. # | VOC | VOC Concentration (ppb) | [NO] (ppb) | Oxidant (ppb) | Notes |
|---|---|---|---|---|---|
| | | Gas-phase experiments | | | |
| G1 | Benzyl Alcohol | 46 | 500 | CH$_3$ONO (300) | |
| G2 | Benzyl alcohol | 60 | 44 | CH$_3$ONO (370) | |
| G3 | Benzyl alcohol | 32 | 0 | H$_2$O$_2$ (2,000) | |
| | | Relative rate experiments | | | |
| G4 | o-Hydroxybenzyl alcohol | na | | na | |
| | | Particle-phase experiments | | | |
| P1 | Hydroxybenzyl alcohol | 191 | 80 | H$_2$O$_2$ (2,000) | |
| P2 | Benzaldehyde | 190 | 14 | H$_2$O$_2$ (2,000) | |



## 2.2 Instrumentation

In the particle-phase experiments, NO and $NO_2$ were monitored using a commercially available Teledyne T200 $NO_x$ monitor. Temperature and RH were measured by a Vaisala HMM211 probe. In gas-phase experiments, $NO_x$ was monitored using commercially available Teledyne $NO_x$ M200 EU. Additional instrumentation used used in gas-phase and particle-phase experiments is described in the following sections.

### 2.2.1 Particle Mass Detection

Particle size distribution was monitored using a custom-built scanning mobility particle sizer (SMPS) which uses a commercially available 308100 TSI Differential Mobility Analyzer (DMA) and a TSI 3010 condensation particle counter (CPC) (Mai, 2018). Prior to the DMA inlet, a soft x-ray charger provided a known charge-distribution. Wall loss corrections and subsequent SOA yields were calculated based on extensive work and model development explained in Charan et al. (2018) and Huang et al. (2018), and detailed more specifically for these experiments in Charan et al. Charan et al. (2020). SOA yields were calculated

based on a ratio of VOC precursor reacted and the formation of SOA:

$$Y = \frac{\Delta SOA}{\Delta VOC \, precursor} \tag{1}$$

Here, $\Delta SOA$ is the change in the aerosol mass concentration while accounting for any aerosol loss to the chamber walls. Aerosol density was assumed to be 1.04 g cm$^{-3}$ based on past work on benzyl alcohol (Charan et al., 2020; Li and Cocker, 2018).

### 95 2.2.2 Gas-Phase Detection

A chemical ionization triple quadrupole mass spectrometer (CIMS) with a $CF_3O^-$ reagent ion was used to monitor gas-phase compounds in particle-phase experiments. This set up has been described in detail elsewhere (Schwantes et al., 2017). In brief, the CIMS operates by reacting with the gas-phase compounds in the sample that have an electron affinity sufficient to bind with the reagent ion cluster ($CF_3O^-$). Sampled compounds that are acidic may also transfer a floride ion ($F^-$). The sample is

then detected by a Varian 1200 triple quadrupole mass analyzer which measures masses from m/z = 50 to 330. A custom-built inlet to the CIMS was set at a constant temperature, 25°C.

Benzaldehyde, one of the primary products of benzyl alcohol OH oxidation, is not detectable using the $CF_3O^-$ CIMS. Thus an HP 6890N gas chromatograph with a flame ionization detector (GC-FID) was used in experiments in Chamber P. Experiments were run with a DB5 column. Information on the temperature profile and GC operation can be found in Appendix

C.

In gas-phase experiments in Chamber G, the GC-CIMS was used to monitor the gas phase precursors and subsequent photooxidation products. This experimental setup is described in detail elsewhere (Vasquez et al., 2018; Xu et al., 2020) Here, the



GC-CIMS was operated in both negative mode using $CF_3O^-$ and in positive mode using $NO^+$. The chemistry involving $CF_3O^-$ is the same as that described previously and in Appendix B. In positive mode, $NO^+$ complexes with less acidic compounds

and can be detected at $[M+NO^+]$ (also described further in B). A 2 m Restek RTX-1701 column was used for all experiments for better chromatographic resolution of certain isomers. GC samples were cryogenically trapped at -20°C on the column. Additional information on the GC-CIMS operation is in Appendix C.

### 2.2.3 Calibrations

Sensitivities of analytes to instrumentation was generally determined using Fourier transform infrared (FTIR) spectroscopy.

The GC-FID and the CIMS in experiments P1 - P2 were calibrated by injecting the analyte of interest into a ~ 100L Teflon pillow bag using the same injection method as described previously for Chamber G. The sample was then measured via a FTIR spectrometer with a pathlength of 19 cm. The reference FTIR spectra from the Pacific Northwest National Laboratory (PNNL) database were used to tabulate cross sections to determine exact concentrations Schwantes et al. (2017). The pillow bag was then diluted using dry $N_2$ and sampled to determine the instrumental sensitivity (Xu et al., 2019).

Several of the substrates involved in this study are relatively non-volatile, leading to challenges with quantitative transfer into and out of the FTIR cell. Therefore some of the GC-CIMS sensitivities were estimated from the measured sensitivity to proxy compounds and using permeation tubes. Additional information on this procedure can be found in SI.2. In short, permeation tubes with the known amounts of the calibrants were connected to the instrument to correlate substrate quantity (weight) and instrument signal.

### 125 2.3 Secondary Chemistry

Oxidation products of benzyl alcohol react as they are formed. These losses were accounted for when calculating branching ratios of benzyl alcohol products. Branching ratios (BR) were calculated as:

$$BR = Y \times CF \tag{2}$$

The gas-phase yield, Y, is calculated as the amount of oxidation product formed divided by the amount of precursor reacted.

To solve for the correction factor, CF, a constant [OH] is assumed for the time-dependent product concentration, which can be described as:

$$[Product]_t = [BA]_0 \times \frac{Y \times k_{BA}}{k_{BA} - k_{Product}} \times [e^{-k_{Product}[OH]t} - e^{-k_{BA}[OH]t}] \tag{3}$$

The time-dependent concentration of benzyl alcohol is described as:

$$[BA]_t = [BA]_0 \times e^{-k_{BA}[OH]t} \tag{4}$$



Therefore CF, the correction factor, is defined as:

$$CF = \frac{k_{BA} - k_{product}}{k_{BA}} \times \frac{1 - [BA]_t/[BA]_0}{([BA]_t/[BA]_0)^{k_{product}/k_{BA}} - ([BA]_t/[BA]_0)} \tag{5}$$

This correction factor is described elsewhere in greater detail (Atkinson et al., 1982). The kinetic rate constants to solve for Eq. 5 can be found in Table 2. For HBA, the correction factor ranged from $\sim 4 - 8\%$, for phenol $< 0.2\%$, benzaldehyde $\sim 0.6 - 2\%$, and hydroxy oxopentenal $\sim 0.6 - 2.9\%$.

| Compound | $k_{OH}$ (cm$^3$molec$^{-1}$s$^{-1}$) |
|---|---|
| Benzyl alcohol | $(2.8 \pm 0.7) \times 10^{-11}$* |
| Benzaldehyde | $(1.29 \pm 0.32) \times 10^{-11}$† |
| $o$-Hydroxybenzyl alcohol | $(4.26 \pm 0.22) \times 10^{-11}$‡ |
| Phenol | $(2.83 \pm 0.57) \times 10^{-11}$†† |
| Butenedial | $(3.45 \pm 0.34) \times 10^{-11}$** |

**Table 2.** Kinetic rate constants used to determine the correction factors for branching ratio calculations. *Harrison and Wells (2009) and Bernard et al. (2013)  †Calvert et al. (2002).  ‡US EPA EPI Suite.  ††Rinke and Zetzsch (1984). ** Martín et al. (2013). Note that for hydroxy oxopentenal, the $k_{OH}$ for butenedial is used to calculate the correction factor.

## 3    Results and Discussion

### 3.1    Gas-Phase Branching Ratios

Gas-phase products were identified in Experiments G1-3 (Table 3 and Figure 1). The oxidation of first generation products were minimized by limiting the reaction of benzyl alcohol to $< 10\%$. The oxidation of benzyl alcohol forms C7 products, such as HBA and benzaldehyde. Additionally, we observed small yields of C6 products, such as phenol and subsequent oxidation

products such as catechol, as well as ring-opening products such as hydroxy oxopentenal. These and other later-generation oxidation products, including fragmentation products, are reported. A list of products and their corresponding CIMS chemistry can be found in Appendix F.

### 3.1.1    Hydroxybenzyl Alcohol and Benzaldehyde

We sought to quantify the yield of the primary benzyl alcohol oxidation products–HBA and benzaldehyde–under low and high

NO conditions. Previous studies of benzyl alcohol oxidation were generally performed under high NO conditions. Thus, we are interested in understanding the extent to which NO affects benzyl alcohol chemistry.





**Table 3.** Results of Gas-Phase Yield Experiments

| Experiment | Hydroxybenzyl Alcohol Yield | Benzaldehyde Yield | Hydroxy Oxopentenal Yield | Phenol Yield |
|---|---|---|---|---|
| 500 ppb NO | $(45 \pm 27)\%$ | $(19 \pm 12)\%$ | $(11 \pm 6.8)\%$ | $(0.66 \pm 0.42)\%$ |
| 44 ppb NO | $(47 \pm 29)\%$ | $(31 \pm 20)\%$ | $(8.0 \pm 4.9)\%$ | $(1.5 \pm 0.96)\%$ |
| 0 ppb NO | $(69 \pm 44)\%$ | $(19 \pm 12)\%$ | $(7.3 \pm 4.6)\%$ | $(1.1 \pm 0.68)\%$ |

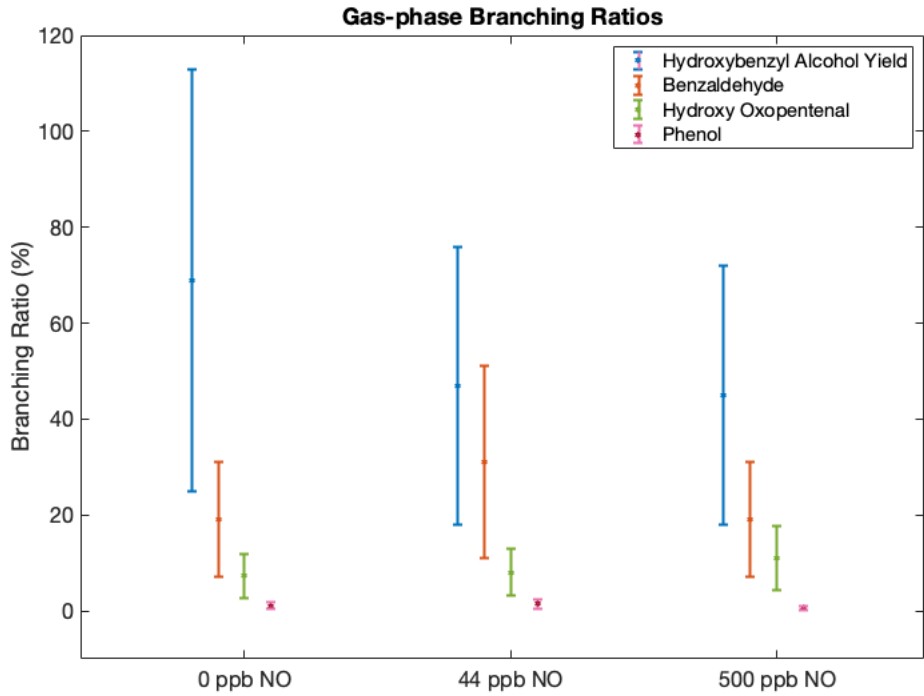

**Figure 1.** NO-dependence of yields for hydroxybenzyl alcohol (HBA), benzaldehyde, hydroxy oxopentenal, and phenol.

While HBA is not currently included in the Master Chemical Mechanism (MCM) scheme, its formation is predicted and identified in past work on benzyl alcohol kinetics and mechanisms (Calvert et al., 2002; Bernard et al., 2013; Wang, 2015; Bloss et al., 2005; Jenkin et al., 2003). HBA forms via the addition of the OH radical to the aromatic ring. This leads to a radical

intermediate which is stabilized by the electron delocalization of the remaining conjugated system. An additional hydrogen abstraction stabilizes the radical intermediate. We calculated the amount of HBA formed from benzyl alcohol oxidation from the gas-phase experiments conducted in Chamber G (Experiments G1-3 in Table 1). We found that the branching ratio of HBA decreases with increasing NO (Figure 1). When [NO] is greater than 44 ppb, we found the HBA branching ratio is ~ $(45 \pm 27)\% - (47 \pm 29)\%$ and in the absence of NO, we found the yield is larger, $(69 \pm 44)\%$. Masses corresponding to later

generation oxidation products of HBA included further OH additions to the aromatic ring, such as dihydroxybenzyl alcohol. We also observed masses that corresponded to fragmentation products observed in past studies or believed to form via theoretical



calculations such as hydroxyoxopropanal. The mechanism for HBA formation and subsequent chemistry can be found in (See Figure 2). An additional summary of compounds detected can be found in Appendix F.

**Figure 2.** Primary reactive pathways for OH-initiated oxidation of benzyl alcohol.

Benzaldehyde forms via initial hydrogen abstraction from the $CH_2OH$ group followed by addition of O2 and subsequent loss
of $HO_2$. Benzaldehyde was measured via $NO^+$ CIMS in Chamber G. The $NO^+$ signal is highly water-dependent and therefore less stable than the $CF_3O^-$ signal. Therefore, it was difficult to confidently quantify the benzaldehyde yield. We estimated the yield is ~ 19% under high and low NO conditions, consistent with the expectation that this channel that should not have NO dependence when NO is less than several ppm. Bernard and coauthors quantified the yield of benzaldehyde to be 25% Bernard et al. (2013). Since benzaldehyde forms via peroxy benzaldehyde, the intermediate may stabilize to other closed shell products
such as hydroperoxide benzaldehyde, and nitrate benzaldehyde. Benzaldehyde can also continue to react via OH addition to the aromatic ring to form products such as hydroxy benzaldehyde.

### 3.1.2 C6 Compounds

A previous study of the chemical composition of the SOA formed via OH oxidation of benzyl alcohol observed C6 compounds, such as nitrocatechol (Charan et al., 2020; Jaoui et al., 2023). In the present study, we observed nitrocatechol as well as other
C6 compounds such as phenol and catechol. Past work has proposed C6 products can form from benzaldehyde oxidation



**Figure 3.** Phenol may form via the HBA radical adduct.

(Figure 3) (Schwantes et al., 2017; Namysl et al., 2020). Further reaction with NO leads to alkoxy benzene which can then stabilize to phenol. Phenol and other C6 products can also form by hydrogen abstraction from the $CH_2OH$ group to form a formaldehyde leaving group. Wang (2015) estimated the barrier to decomposition is too high for significant yields of phenol, though predicted low yields of phenol via decomposition of the HBA radical adduct to form phenol and methanol. In the present study we observed small initial yields of phenol $< 1\%$. We observed that the yield of phenol decreases with decreasing NO, consistent with past studies showing the mechanism for phenol formation depends on NO mixing ratio (Xu et al., 2020).

We observed C6 compounds in both gas-phase and particle-phase experiments. Phenol and other C6 aromatic compounds react rapidly with OH to form other oxygenated aromatic compounds ($\leq 2.7 \times 10^{-11}$ cm$^3$molecule$^{-1}$s$^{-1}$) (Calvert et al., 2002). Thus, despite relatively low yields of phenol in the first generation of this chemistry, we observed many C6 products in the aerosol. Wang (2015) also predicts a considerable yield of nitroaromatics. In high NO conditions, we observed nitrophenol in gas-phase and particle-phase experiments. Furthermore, in past work, nitrocatechol is identified as an important compound in the chemical composition of aerosol formed from benzyl alcohol oxidation (Charan et al., 2020).

### 3.1.3  Fragmentation and Ring-Opening Products

Notable fragmentation products include hydroxy oxopentenal and butenedial, the latter of which is also reported elsewhere (Harrison and Wells, 2009; Birdsall et al., 2010). We expect a bicyclic peroxy intermediate forms, congruent with the chemistry of other aromatic systems, which leads to, among other products, ring-opening and fragmentation compounds (Calvert et al., 2002). In higher NO conditions, the bicyclic peroxy intermediate goes on to form a bicyclic alkoxy intermediate which eventually fragments to form products such as hydroxy oxopentenal. Thus, as NO increases, the yield of of hydroxy oxopentenal also increases. Nonetheless, though, at low NO conditions, fragmentation products still formed in significant quantities possibly because alkoxy formation may occur from the reaction of $HO_2$ with the precursor $RO_2$ (Table 3). This behavior is congruent with that of other aromatic systems, such as toluene, in which increased NO favors formation of this ring-opening products, in part due to the formation of a bicyclic peroxy radical (Figure 4) (Birdsall et al., 2010). Notably, in other aromatic systems where the bicyclic intermediate forms, so too can a fragmented epoxide form (Figure 4) (Birdsall et al., 2010). As found in other systems, epoxides can effectively lead to SOA formation (Paulot et al., 2009; Chan et al., 2010). Other fragmentation products found in Jaoui et al. also suggest that fragmentation products can form with high O:C ratios and may quickly partition to SOA (Jaoui et al., 2023).





**Figure 4.** Following addition of OH to the aromatic ring, a bicyclic intermediate can form which can eventually lead to form fragmentation products. Here, we detect products with masses congruent with both the hydroxy oxopentenal and epoxide products.

### 3.2 Particle-Phase Results

Because HBA has a much lower vapor pressure than benzaldehyde, HBA likely contributed more to SOA formation in benzyl alcohol oxidation than benzaldehyde. This hypothesis was tested by conducting SOA experiments using HBA and benzalde-

hyde as the VOC precursors (P1 – 2). We estimate the relative contributions of HBA and benzaldehyde to benzyl alcohol SOA formation by conducting individual SOA yield experiments using HBA and benzaldehyde as the VOC precursors. Conditions for SOA experiments were selected to match those of the benzyl alcohol SOA yield experiments in Charan et al. (2020). In brief, experiments were conducted with ~ 80 ppb of initial NO, ~ 80 ppb of VOC precursor, and ~ $1 \times 10^4$ cm$^{-3}$ of inorganic seed aerosol. SOA yield results were calculated using two treatments: one where the proportionality factor, $\omega$, was set to unity

and another in which $\omega$=0. In the $\omega$=0 case, oxidation products with a sufficiently low vapor pressure to condense were assumed to do so only on suspended particles and not on particles that had deposited on the chamber walls (Weitkamp et al., 2007). When $\omega$=1, on the other hand, condensable oxidation products and particles deposited on the chamber walls during the experiment were assumed to be in equilibrium with one another (Weitkamp et al., 2007).

Figure 5 shows results for SOA yield experiments using benzaldehyde and HBA. Note the OH exposures for benzaldehyde

and HBA in these experiments were ~ $2.8 \times 10^{10}$ molecule s cm$^{-3}$ and ~ $2.8 \times 10^{11}$ molecule s cm$^{-3}$, respectively, which corresponds to ~ 52 min - 8.6 h of OH exposure on a typical Los Angeles summer day (Appendix H) (Griffith et al., 2016). Both experiments were allowed to react for the same amount of time, ~ 400 minutes, to compare to the SOA yield experiments conducted in Charan et al. (2020). Therefore, we report the SOA yields of HBA and benzaldehyde at t = 400 mins (SOAY$_{400mins}$), rather than at an equilibrium point. In the upper bound, ($\omega = 1$), SOAY$_{400mins}$ of HBA is $1.2 \pm 0.14$ and in the





lower bound ($\omega = 0$), $0.97 \pm 0.13$. For benzaldehyde, we report an upper bound of $\text{SOAY}_{400\text{mins}}$ $0.35 \pm 0.039$ and a lower bound

of $0.28 \pm 0.039$. While comparing $\text{SOAY}_{400\text{mins}}$ is helpful in determining which pathways (addition versus abstraction) con-

tribute to the high SOA yield of benzyl alcohol, this value does not necessarily inform atmospherically relevant SOA yields of

benzaldehyde and HBA as inputs for models because modeled reaction time and conditions may not exactly match experimen-

tal ones. Therefore, we report the parameterized fit of the SOA yields as a function of absorbing organic mass concentration

(M).

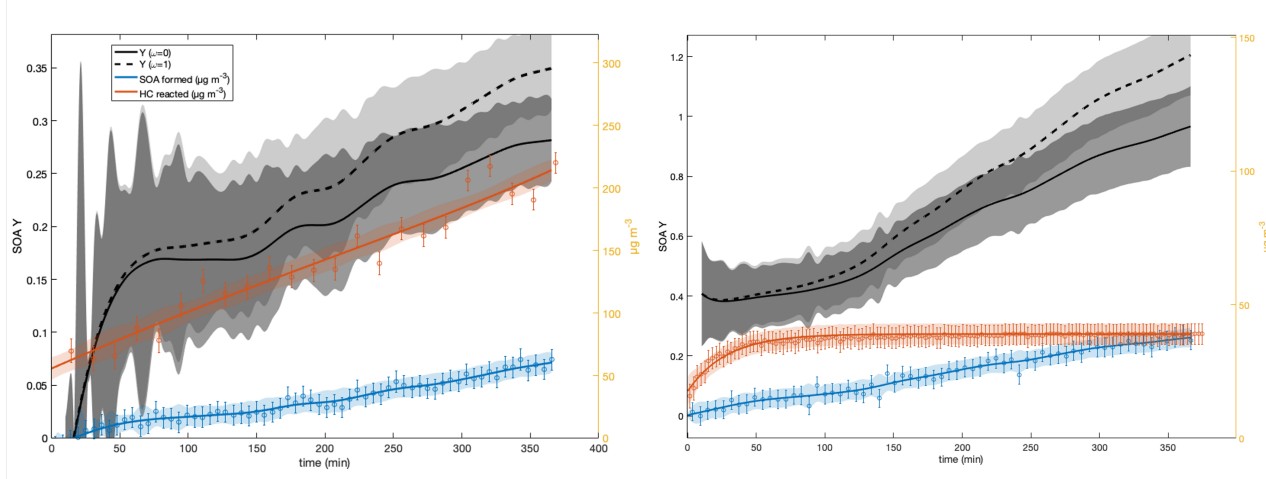

**Figure 5.** Wall-loss corrected SOA yields of benzaldehyde (a) and HBA (b). Solid yields are calculated assuming $\omega$ is zero. Dotted yields are calculated assuming $\omega$ equals unity. Red data are the amount of VOC precursor reacted in $\mu g \, m^{-3}$. Data displayed in blue are SOA formed in $\mu g \, m^{-3}$.

We follow a one-product parameterization method that follows the multiple paramterization described in Odum et al. Odum et al. (1996) where,

$$Y = M \times \left( \frac{\alpha K_{om}}{1 + K_{om} \times M} \right) \tag{6}$$

Here $\alpha$, $K_{om}$ is the partitioning coefficient and where $\alpha$ is a constant relating the total concentration of products formed with

the amount of organic gas-phase mass reacted. The parameters in Equation 6 were chosen by minimizing the least square fit to

the data (Appendix E). Results for benzaldehyde SOA yield are graphed in Figure 6 ($\alpha = 0.34 \pm 0.013$ and $K_{om} = (9.4 \pm 1.2) \times$

$10^{-3}$). For HBA, this approach becomes more complicated. The Odum et al. approach uses the steady state approximation

(SSA) which states that the derivative of the concentration of an intermediate species appears to be zero. In other words, we

assumed an approximately steady state of first-generation oxidation products which are reacting at roughly the same rate as

their formation. The rates of reaction of HBA and subsequent oxidation products were likely unequal because the HBA reacts



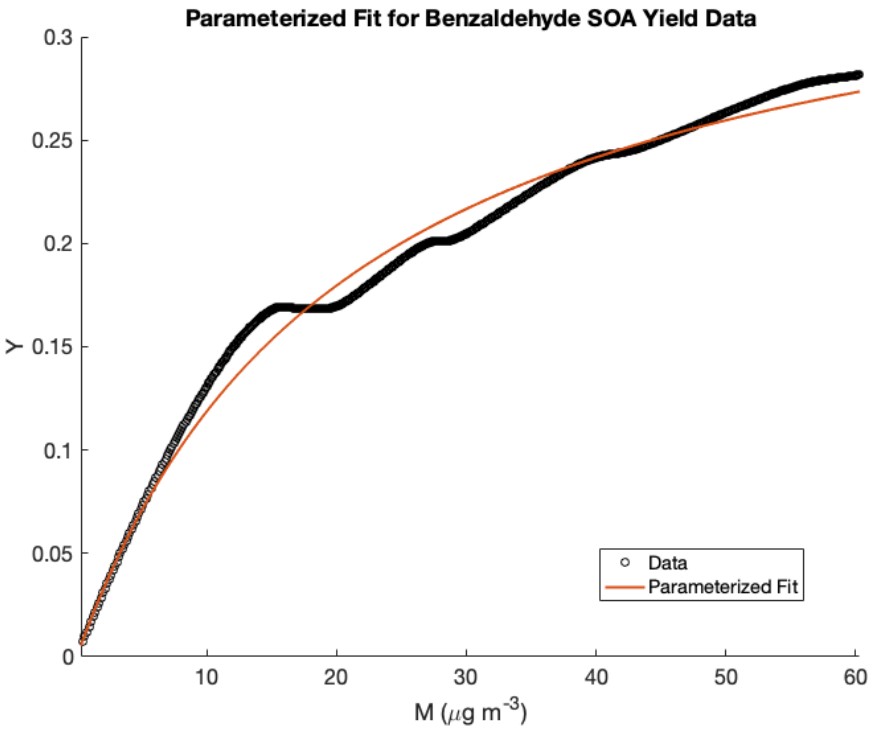

**Figure 6.** Parameterization of SOA yield data as a function of organic mass reacted. Because SOA yields did not stabilize in these experiments, parameterization can be useful in contextualizing SOA yields under atmospherically relevant conditions.

away early in the experiment; therefore, the SSA was not a sufficient approximation and so, we did not use the Odum et al. fitting for HBA.

At t = 400 mins, unreacted benzaldehyde remained in the chamber whereas HBA reacted within the t < 1 h of the experiment. This may indicate that the SOA formed in the benzaldehyde experiments is from the very rapid chemistry of subsequent
generation oxidation products, as is the case with toluene. In the HBA experiments, most of the SOA is likely also generated from the oxidation of later-generation products.

Because HBA had a much larger SOA yield than benzaldehyde and a large branching ratio from benzyl alcohol oxidation, HBA likely contributed most significantly to the large measured SOA yield of benzyl alcohol in our experiments. We estimate, by competitive kinetics, that the reaction of HBA + OH proceeds at $(2 \pm 0.1) \times 10^{-11}$ cm$^3$molecule$^{-1}$s$^{-1}$ (Appendix D). The
rapid reaction of HBA likely leads to other products with increasing OH kinetics. In aromatic oxidation chemistry, addition of electron-donating groups (such as OH) lowers the barrier of reaction for additional OH chemistry (Calvert et al., 2002). With benzene, adding electron-donating substituent groups can increases reactivity with OH by ~ 4 − 8x. In the benzyl alcohol system, after every subsequent reaction with the OH radical, we anticipate the kinetics of HBA quickly lead to low-volatility, highly-oxygenated products that readily partition into aerosol.



HBA predominately contributing to SOA further agrees with past SOA studies of benzyl alcohol conducted as a function of NO. In Charan et al. (2020), SOA yields of benzyl alcohol increased with decreasing NO. Similarly, here we observed that in gas-phase experiments (G1 - 3), the HBA yield increased with decreasing NO. Because HBA primarily contributed to the high SOA yield of benzyl alcohol, it follows that as the gas-phase yield of HBA increased, the SOA yield of benzyl alcohol also increased.

## 3.3   Comparison With Past Work

Past studies have detected HBA and benzaldehyde from the oxidation of benzyl alcohol (Wang, 2015; Bernard et al., 2013; Harrison and Wells, 2009). Wang (2015) estimates the yield of o-HBA is $11\%$ using theoretical estimates, while Bernard and coworkers estimate the yield is $21\%$; at similar [NO] we found the branching of HBA is ~ $45\%$. We observed the HBA branching ratio increases with decreasing NO. In the absence of NO we found the yield is ~ $69\%$. Variations in yields are

likely due to differences in initial conditions. Though [NO] may have been similar across different works, yields are also affected by [$O_2$], the initial benzyl alcohol mixing ratio, amount of benzyl alcohol reacted, OH exposure, and temperature, among other variables. In Bernard et al. (2013) starting benzyl alcohol mixing ratios ranged from ~ 2000 ppb - 850 ppb, whereas in our system initial [benzyl alcohol] < 100 ppb. If the system is allowed to react for sufficiently long, secondary chemistry can play an increasingly significant role. Unaccounted for secondary reactions may lead to an under-reporting of the

HBA branching ratio, especially in longer oxidation experiments. Additionally, to our knowledge, no yield studies have been previously conducted for this system in the absence of NO.

We observed only one isomer of HBA which we assume to be the *ortho* product. This assignment is based on past work that suggests the *ortho* position is a major product in aromatic oxidation chemistry by OH (Harrison and Wells, 2009; Finlayson-Pitts, Barbara and Pitts, James N. Jr., 1986; Baltaretu et al., 2009). Similarly, Wang (2015) predicted a single stable isomer of

HBA, *ortho*. If additional HBA isomers existed in our system, it is likely they would have eluted at higher temperatures than were allowed by the current GC temperature profiles and were therefore undetected. However, secondary isomer formation is typically considered to be minor in other aromatic systems (Finlayson-Pitts, Barbara and Pitts, James N. Jr., 1986; Baltaretu et al., 2009).

Bernard et al. (2013) also reported a benzaldehyde yield of $25\%$, which was used as input to the computations performed in

Wang 2015, while Harrison and Wells report a yield of $24\%$ (Bernard et al., 2013; Wang, 2015; Harrison and Wells, 2009). We report a benzladehyde yield of ~ $19\%$, in close agreement with past work. Closer agreement between our work and past work may be because benzaldehyde oxidation is not dependent on [NO], unlike HBA.

## 4   Conclusions

Benzyl alcohol oxidizes via OH to primarily form HBA and benzaldehyde. Significant additional chemistry occurred via

endocylization following addition of oxygen to fragmentation products such as hydroxy oxpentenal and butadiene. We found that [NO] has an effect on several of the product yields including HBA and hydroxy oxopentenal, though the overall relative



distribution of oxidation products remained unchanged–namely that HBA was the dominant first-generation product with a branching ratio of $69-45\%$ over a range of NO conditions. The branching ratio of benzaldehyde did not appear to be [NO]-dependent and is estimated to be ~ $19\%$.

Both HBA and benzaldehyde pathways go on to form highly oxygenated gas-phase products. HBA oxidation leads to the formation of products such as dihydroxybenzyl alcohol, while similarly, benzaldehyde oxidation forms products such as dihydroxy benzoic acid. These products indicate that subsequent OH addition to the aromatic ring occured in both pathways. Both the addition and hydrogen abstraction routes may have also contribute to the formation of C6 products.

Aerosol yield studies using HBA and benzaldehyde as the precursors suggested that the HBA pathway is important to the 290 high SOA yields observed in benzyl alcohol oxidation. HBA is quickly oxidized by OH ($k_{OH} = (2\pm0.1)\times10^{-11}$ cm$^3$molecule$^{-1}$s$^{-1}$) to form low-volatility products which rapidly partition to the particle phase, thus contributing to the high SOA yield of benzyl alcohol. Though VCPs have been identified as increasingly important to SOA formation, key VOC components of VCPs remain uncharacterized. Products from benzyl alcohol oxidation via OH were identified here, elucidating its fast reactivity and high aerosol mass yields.

*Data availability.* Data from this study can be made available upon request. Data for experiments P1 and P2 can be found through the Integrated Chamber Atmospheric Data Repository for Unified Science (ICARUS) and upon request.

**Appendix A: Experimental Conditions**

CH$_3$ONO (synthesized following Taylor et al. (1980)), and NO (1993±20 ppmv, Matheson) were injected into the Chamber G (~ 800 L) in a similar fashion. The analyte is introduced to an evacuated 0.5 L glass bulb and is serially diluted with N$_2$ 300 until the desired mixing ratio is achieved. CH$_3$ONO was quantified via FTIR spectroscopy using tabulated cross section prior to being injected into Chamber G. Ultraviolet lights (Sylvania) centered around 350 nm were used. The measured j$_{CH3ONO}$ and inferred j$_{NO2}$ from these lights are $1.1\times10^{-3}s^{-1}$ and $4.4\times10^{-3}s^{-1}$, respectively.

**Appendix B: CIMS Calibration and Instrument Sensitivity**

The CIMS operates by flowing the reagent ion source (CF$_3$OOCF$_3$ in negative mode and NO·H$_2$O$^+$ in positive mode) through 305 a radioactive source ($^{210}$Po), generating the reagent ions. In negative mode, cluster ions (Eq. B1) or transfer ions (Eq. B2) are produced. CF$_3$O$^-$ CIMS chemistry is documented extensively in past studies (Vasquez et al., 2018; St. Clair et al., 2010; Crounse et al., 2006). In short, the CF$_3$O$^-$ ion is sensitive to a variety of atmospherically-relevant oxygenated species.

$$M + CF_3O \leftrightarrow M \cdot CF_3O^- \tag{B1}$$



$$\mathrm{M + CF_3O \rightarrow M-H^- \cdot HF + CF_2O} \tag{B2}$$


In positive mode, $\mathrm{NO \cdot H_2O^+}$ is produced (Eq. B3). Ambient $\mathrm{N_2}$ and $\mathrm{H_2O}$ react with NO. The $\mathrm{NO \cdot H_2O^+}$ tends to bind to less oxygenated species including carbonyls such as benzaldehyde.

$$\mathrm{N_2^+ + NO \rightarrow NO^+ + N_2} \tag{B3}$$

$$\mathrm{NO^+ + H_2O^+ \leftrightarrow M \cdot NO^+ + H_2O} \tag{B4}$$


The CIMS in experiments in Chamber P was calibrated for benzyl alcohol and benzaldehyde. The GC-CIMS in experiments in Chamber G was calibrated for phenol and benzaldehyde. Individual calibrants were injected into a ~ 100 L Teflon pillow bag using the same injection method as described previously. The sample was then measured via a Fourier transform infrared (FTIR) spectrometer with a path length of 19 cm. The reference FTIR spectrum from the Pacific Northwest National Laboratory
(PNNL) database was used to tabulate cross sections to determine the exact concentration of the calibrant (Schwantes et al., 2017). The pillow bag was then diluted using dry $\mathrm{N_2}$ and sampled to determine the instrumental sensitivity. The GC-FID was calibrated using benzaldehyde and a similar procedure. In some cases, an averaged sensitivity was used for compounds as described in Xu et al. (2019).

For hydroxy oxopentenal, glycolaldehyde was used a proxy, as detailed in Yu et al. Yu, Hongmin et al. (2023). In short,
calculated dipole moments and polarizability of glycolaldehyde (at B3LYP/cc-pVTZ level). These calculations are then used to determine the compound-specific sensitivity to the CIMS signal. Glycolaldehyde was chosen as a proxy compound for hydroxy oxopentenal because of their structural similarities and thus their similar chemistry with the reagent ion.

For hydroxybenzyl alcohol (HBA), catechol was used as a calibrant proxy relative to phenol. Catechol was selected as a proxy for HBA because of their similarities in structure, namely the two hydroxyl groups. Based on this functionalization, both
HBA and catechol primarily bind with the transfer ion, $\mathrm{F^-}$.

Benzyl alcohol and phenol were calibrated directly on the CIMS for particle-phase experiments using the method describe in the Calibration section of the main paper. These calibrations were then used to estimate the sensitivity of benzyl alcohol for the GC-CIMS used in gas-phase experiments. These calculations were performed based on the assumption that the same yield of phenol will form from benzyl alcohol under similar conditions (e.g., initial [NO]).

Catechol and phenol were calibrated using permeation tubes which were weighed to determine the exact mass of analyte. This known amount substrate was then correlated to signal on the GC-CIMS.



## Appendix C:  GC Operation

In gas-phase experiments, analyte samples were cryogenically trapped in the GC-CIMS at ~ −20°C for 10 mins using liquid $CO_2$. The sample was eluted through the GC using a ramp of 10°C/min to 55°C and then 2.5°C/min to 130°C. The slower ramp
rate from $55 - 130°C$ was used because most oxidation products eluted at this time and using a slower ramp ensured relevant products were sufficiently separated. When GC scans were not being taken, the CIMS sampled directly from the reaction chamber. For particle-phase experiments, the GC-FID was run from 40°C to 250° C with a ramp rate of 50°C/min. A DB-5 column was used for these experiments.

## Appendix D:  The Reaction Rate Coefficient for OH + HBA

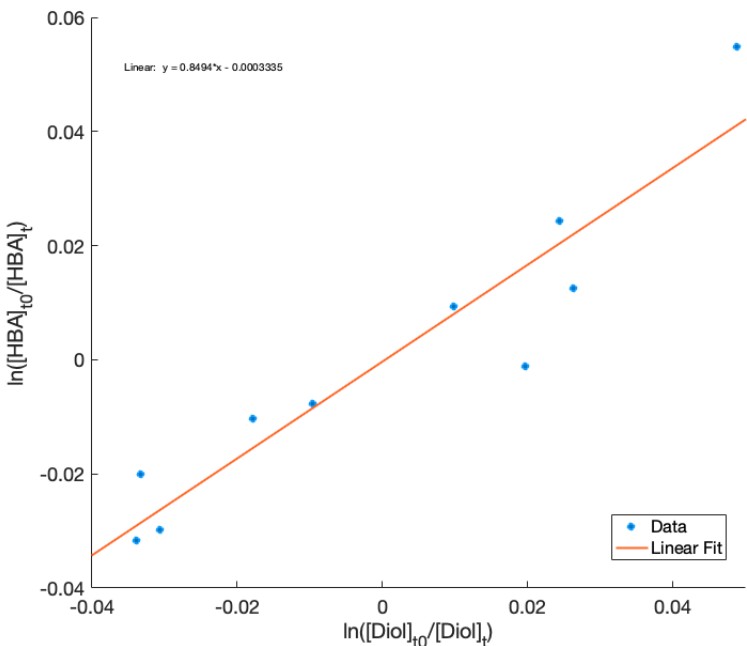

**Figure D1.** Relative rate of reaction of HBA compared to butanediol.

Relative rate experiments of HBA were conducted using butanediol as a reference. The assumed kinetic rate constant for butanediol was $(2.7 \pm 0.48) \times 10^{-11} cm^3 molecule^{-1} s^{-1}$ Bethel et al. (2001). Methods to calculate $k_{OH}$ are detailed in Bethel et al. (2001) Bethel et al. (2001). In brief, ~ 20 ppb of 1,2-butanediol was injected into Chamber G using the procedure described in Methods. Similar amounts of HBA were also injected. Methyl nitrate (~ 500 ppb) was used as the oxidant source and ~ 333 ppb of NO (Matheson, $1993 \pm 20$ ppb) was also injected. Oxidation was initiated using 8 UV lights (Sylvania) centered around
350 nm lights. Short periods of oxidation (~ 1 min) were followed by periods without irradiation to allow the CIMS signal to



stabilize. This proceeded until approximately one e-fold of VOC was reacted. The plotted natural log quotients of initial and final mixing ratios of butanediol and HBA can be found in Figure D1. The slope of this plot provides a ratio of their respective kinetic rate constants:

$$slope = \frac{k_{HBA}}{k_{diol}} \tag{D1}$$

The slope of the natural log quotients can be found in Figure D1. Based on these data, the relative rate of HBA is $(2 \pm 0.1) \times 10^{-11} \text{cm}^3 \text{molecule}^{-1} \text{s}^{-1}$.

**Appendix E:  Parameterization of SOA Yields**

We report the results of the parameters from the least square fit for benzaldehyde derived SOA. For Eq. 6, $\alpha = 0.34 \pm 0.013$ and $K_{om} = (9.4 \pm 1.2) \times 10^{-3}$. The fit has an r-squared value of $0.90$.

**Appendix F:  Oxidation Products Detected**

The following table summarizes masses detected by the GC-CIMS in gas-phase experiments (G1 - G3). While some products were identified using authentic standards, other assignments are based on masses detected and our chemical understanding of this and other aromatic systems.

**Appendix G:  Estimation of Uncertainty**

In gas phase experiments, the major source of error in our calculation of yields comes from the relative sensitivity of the product concentration as measured by their respective instruments. A part of the reported error is also determined by calculating the standard deviation of signal variation. This varies from compound to compound and is also dependent on the reagent gas used. Uncertainty in quantifying the relative signals in positive and negative. Based on this, we estimate the error for compounds detected by the GC-CIMS is $\sim 63\%$. Phenol was directly calibrated for branching ratio calculations, therefore the estimated
uncertainty for phenol is $21\%$.

**Appendix H:  OH Exposure**

[OH] was calculated for SOA yield experiments using the kinetic equation:

$$\frac{d[\text{reagent}]}{dt} = -k_{OH}[\text{OH}][\text{reagent}] \tag{H1}$$



| VOC Compound (m/z) | Structure | Formula | Observed m/z and Reagent Ion |
|---|---|---|---|
| Benzyl alcohol (108) | | $C_6H_5CH_2OH$ | 193 ($CF_3O^-$) |
| Benzaldehyde (106) | | $C_6H_5C(O)$ | 136 ($NO^+$) |
| Hydroxybenzyl alcohol (124) | | $C_6H_4OHCH_2OH$ | 143 ($F^-$) and 209 ($CF_3O^-$) |
| Dihydroxybenzyl alcohol (140) | | $C_6H_4OHOHCH_2OH$ | 225 ($CF_3O^-$) |
| Phenol (94) | | $C_6H_5OH$ | 113 ($F^-$) and 179 ($CF_3O^-$) |
| Catechol (110) | | $C_6H_4OHOH$ | 129 ($F^-$) and 195 ($CF_3O^-$) |
| Hydroxymethylbutenedial (114) | | $C_5O_3H_6$ | 199 ($CF_3O^-$) |
| Nitrophenol (139) | | $C_6H_5NO_3$ | 158 ($F^-$) |
| Dihydroxybenzoic acid (154) | | $C_6H_3OHOHC(O)H$ | 173 ($F^-$) |
| Bicyclic aromatic (219) | | $C_6O_7NH_9$ | 173 ($F^-$) |
| Tetrahydroxy benzene (192) | | $C_6H_6O_4$ | 211 ($F^-$) |
| Glyoxal (58) | | $C_2H_2O_2$ | 99 ($NO^+$) |
| Formic acid (46) | | $CH_2O_2$ | 86 ($NO^+$) |
| Benzoquinone (108) | | $C_6H_4O_2$ | 148 ($NO^+$) |
| Oxopentanal (100) | | $C_5H_8O_2$ | 140 ($NO^+$) |
| Butenedial (84) | | $C_4H_4O_2$ | 124 ($NO^+$) |
| Hydroxyoxopropenal (88) | | $C_3H_4O_3$ | 173 ($CF_3O^-$) |
| Oxopropanoic acid (88) | | $C_3O_2H_4$ | 173 ($NO^+$) |
| Methanol butenedial (102) | | $C_5O_5H_7$ | 187 ($CF_3O^-$) |
| Hydroxyacetaldehyde (60) | | $C_2O_2H_4$ | 145 ($CF_3O_0$) |

**Table F1.** Compound assignments from CIMS data. Note that several of the compounds listed have many isomeric structures though only one may be listed as an example.



Here, $\frac{d[\text{reagent}]}{dt}$ was determined by finding the slope of the starting reagent against time. For benzaldehyde, the slope over the entire experiment was used. For HBA, the slope for the first 15 min was used to determine [OH] for the experiment.

*Author contributions.* RSB: study conceptualization, data collection and analysis, result interpretation, and writing. SMC: Analysis code and writing. JHS: supervision and writing. POW: supervision, result interpretation, and writing.

*Competing interests.* The authors have no competing interests to declare.

*Acknowledgements.* This research was supported by the National Science Foundation (grant no. CHE-2305204) and the Alfred P. Sloan Foundation (grant no. G-2019-12281). Work performed by RSB was supported by the National Science Foundation Graduate Research Fellowship (grant no. 1745301). The authors thank Yuanlong Huang for help troubleshooting instrumentation in the lab and Lu Xu and Katharine Ball for their work on phenol and catechol calibrations.



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
