# Peer review of "Quantifying primary oxidation products in the OH-initiated reaction of benzyl alcohol"

_EGUsphere, 2023_

## Author Comment (AC1)

We thank the reviewers for their helpful comments on the manuscript. The lengthy delay in reply reflected additional laboratory work to firm up the quantification of the first-generation oxidation products of benzyl alcohol.

**Responses to Referee 1**

We thank reviewer 1 for their time and have updated the manuscript to address their concerns. In size 12pt Calibri we address the individual comments or reviewer 1.

Overall resolution of the figures seems inconsistent. Figures 2 and 4 should be replaced with high-resolution versions. Figure 5 has a gray border that should be removed, and the font size should be increased, to at least match that of Figures 6 and D1.

Figures 2 and 4 were replaced with higher resolution versions.

Line 68: Check the product number of the TSI soft x-ray charge conditioner, is it 2088 or 3088?

Line 86: Check the product number of the DMA, I assume it is 3081 not 308100.

Product numbers for the TSI soft x-ray charge conditioner and DMA were updated.

Line 123: Reference to SI (no SI is available), should be Appendix B.

References to SI updated to reference relevant appendixes.

Line 208: The concentration of NO in the "particle phase experiments" mentioned to be ~80 ppb. In Table 1, the HBA experiment indeed uses 80 ppb, but this is not the case for the benzaldehyde experiment ([NO] = 14 ppb) which is inconsistent with the language used in line 208.

Thank you for catching this. Corrected $NO_x$ reported in the experimental summary.

Line 347: Bethel et al. is mentioned repeatedly.

References were corrected in Line 347.

Line 350: "350 nm lights."  Delete "lights".

Made correction.

Table F1: Numbers in the "Observed m/z and Reagent Ion" column, particularly after Glyoxal, seem inconsistent. Glyoxal + NO+ should be 88, not 99, for example, among others. The very last $CF_3O^0$ should be corrected to $CF_3O^-$.

Information in Table F1 was corrected: The molecular formula for dihydroxy benzoic acid was updated (changed from C6H3OHOHC(O)H to C6H3OHOHC(O)OH).

Tetrahydroxy, benzoquinone, oxopentanal, MW information was corrected.

Corrected methanol butenedial molar mass and molecular formula

Corrected hydroyoxopropanal nomenclature and structure.

---

## Author Comment (AC2)

We thank the reviewers – especially reviewer 2 – for their helpful comments on the manuscript. The lengthy delay in reply reflected additional laboratory work to firm up the quantification of the first-generation oxidation products of benzyl alcohol. The new experiments are described in the revised manuscript and are helpful in addressing the concerns of reviewer 2.

**Response to Reviewer 2**

We thank reviewer 2 for their time and helpful comments. We have updated the manuscript in many places to address their concerns. In size 12pt Calibri we address the individual comments for reviewer 2.

In the SOA experiments 850 µg/m3 of HBA were added to the chamber but only 40 µg/m3 reacted (Table 1, Figure 5). There is no reason why not more HBA should react after 50 min reaction time.

Made corrections to experimental table. We also modified the SOA wall-loss fits for the data presented which modified the end SOA yields. Information was added to the wall-loss appendix to include additional details about the particle wall-loss treatment.

It is also not clear, why HCreact does not start at zero in the beginning of the experiment in both cases.

Thank you for catching this detail. For HBA, the time was offset so it made it look like the zero point was offset. We've zoomed out so that the time axis is set correctly. For the benzaldehyde plot, we've made corrections to the plot. To calculate the deltaHC, we average over a period prior to irradiation to get an accurate initial [HC]. In this case, we included data prior to irradiation and prior to when there was a stable signal which gave a higher $[HC]_0$ than was accurate, therefore making $[HC]_{1\ min}$ look << $[HC]_0$.

There is also SOA available at the beginning of the HBA experiment or the y-axis scale is wrong. Could it be that HBA is lost to the wall and there is an equilibrium between wall and gas phase such that any further reaction of HBA cannot be observed? This would mean that not only the OH exposure would be wrong, but also the SOA-yields. From these experiments as presented here one cannot conclude anything.

Regarding there being "SOA available at the beginning of the HBA experiment": We understand the confusion with Figure 5 and have labeled it better to avoid further confusion. The SOA plot starts slightly after the experiment starts. This is because there is low sensitivity at the beginning of the experiment, so it is difficult to calculate SOA accurately at the start. However, you can see that the plot for just aerosol starts at zero at the beginning of the experiment. This method for displaying the data has been published previously (e.g., see Charan et al. 2020 also in ACP)

Regarding the gas phase experiments the authors report only one experiment for each of the 3 NO levels. The experiment without NO has a yield of (69 ± 44)%, which overlaps largely with the uncertainty range of the other experiments. The uncertainty of the no-

NO experiment is also much larger than for the others. How can the authors be so confident that HBA-yield is much higher in the no-NO experiment, which is the main result of this paper?

We conducted additional experiments at similar NO mixing ratios. In these new experiments we added a GC-FID instrument to measure benzaldehyde. We also changed our approach to calibrations. Most analytes are now calibrated based on computed ion-molecule collision rates. For HBA, we now use a structure additivity approximation based on the kinetics of cresol, which has a similar structure to HBA. See changes in Table 3 and throughout manuscript.

In Table 3 even not all first generation ring opening products are included as well as compounds that could not be detected by their method. This could indicate that their measured yields could be strongly biased high. It is also known that compounds like benzylalcohol and its oxidation products may be prone to wall loss. How did the authors correct for this?

- In Table 3 in the results section, we only include compounds that were both detectable and quantifiable using both negative mode ($CF_3O^-$) and positive mode ($NO^+$) with our CIMS. For additional oxidation products and fragmentation products, please see Figures 2, 3, and 4. We note that there are likely other unmeasured products formed via endocylization chemistry.
- Regarding gas-phase wall loss: in our new experiments we took samples before and after oxidation to examine the stability of the CIMS and GC-FID signals with no oxidation. We have included in the manuscript that we find stable CIMS and GC-FID signals for benzyl alcohol and quantified oxidation products over the time periods used for calculations. This appears congruent with past work on quantifying benzyl alcohol products, as well as other work quantifying first-generation gas-phase products typically consider gas-phase wall loss to be negligible. Please see Charan et al. (2020) in ACP, Jaoui et al. (2023) in ACP, Bernard et al. (2013), and Harrison and Wells (2009) as examples.

While the benzaldehyde yield is similar to other studies, HBA yield is much higher. Could this also be an issue of calibration. As I understand the uncertainties also include a potentially systematic shift, i.e. if the sensitivity with respect to the used proxy differs in one or the other direction. Thus, the HBA yield could be systematically lower or higher. For example in the case of no-NO, the yield of all compounds given in Table 3 ranges from 35% to 158%.

We have updated the branching fractions using an improved approach to calibrations based on calculated dipole moments and polarities of each analyte. We believe that the yield is quite high; differences with previous measurement of the branching fraction may reflect similarly the challenges in both quantifying this 'sticky' compound and accounting for its fast secondary chemistry.

Eq. 2: this equation is a bit confusing. You use branching ratio and yield interchangeably. Branching ratios are usually ratios of two reaction paths and not

fractions as it should be in this context. In Table 3 yields are reported and not branching ratios.

Apologies for the confusion. We have changed all instances of "branching ratios" to "branching fraction" to make it clearer that we are attempting to quantify the fraction of the total oxidation going down specific pathways. To estimate the branching fractions, we take the observed molar yields and correct for secondary losses, as denoted in Equation 2.

Line 145: not all ring-opening products are later generation products, see Figure 4.

This was not the intent of this sentence. Rephrased for clarity.

Figure 1: for consistency I suggest to use yields for axis title and figure title. The numbers shown are fractions and not ratios.

Figure 1 was deleted.

Figure 2: In the phenol path only CH2OH not methanol can leave, to form phenol. In the benzaldhyde path OH is not removed. HO2 is formed from H-abstraction. In the HBA path OH does not add but oxygen abstracts H-atom, or oxygen adds and HO2 leaves.

Note that Figure 2 is now Figure 1. Thank you for catching these errors. Phenol and benzaldehyde paths corrected. $HO_2$ added to H-abstraction benzaldehyde path. We also corrected the HBA path.

Line 167: it says that yield of benzaldehyde channel should not be NO-dependent. Give reference.

Similar to the chemistry following the abstraction of hydroperoxyl hydrogen in HMHP, we expect that, following the addition of $O_2$ alpha to the OH group in benzyl alcohol, the rate of $HO_2$ loss will exceed that of the bimolecular reaction of the RO2 with NO even at the highest NO concentration studied here. In the HMHP system, Allen et al. observed that the formation of formaldehyde from HMHP did not depend on NO for [NO]= $3 \times 10^{10}$ - $1.5 \times 10^{13}$ molecules $cm^{-3}$ (Allen, H. et al., *J. Phys. Chem A* (2019). https://doi.org/10.1021/acs.jpca.8b04577)

Line 169: peroxy benzaldehyde: how does this radical look like? I do not think this is the correct name. Is it not something like a-peroxy-benzylalcohol?

Corrected the description of benzaldehyde chemistry.

Line 170: formation of hydroperoxide benzaldehyde would mean that an H-atom leaves the intermediate instead of HO2. Is this a feasible pathway? How can nitrate benzaldehyde form?

The hydroperoxide is not adding to the aromatic, rather we are referencing the peroxyacetyl radical. To clarify, we have changed the wording in the manuscript to reflect the correct nomenclature. We have Benzaldehyde + OH + $O_2$ would lead to a peroxy radical (on the aldehyde moiety) which can react with $HO_2$ to form a peracid (and likely other products) or

react with NO$_2$ to form peracetyl nitrate benzaldehyde [*The Mechanisms of Atmospheric Oxidation of Aromatic Hydrocarbons* by Calvert, Atkinson, Becker, Kamens, Seinfeld, Wallington, and Yarwood].

Line 176: Figure 3 does not show benzaldehyde oxidation, rather benzylalcohol oxidation. Furthermore, Schwantes does not show phenol formation from alkoxy benzene, it is nitrophenol or nitrosophenol. The mechanism given in Namysl is at high temperature and does not apply here.

Schwantes et al. does explicitly show formation of C6 compounds from benzaldehyde (for example, see Figure 7 of the cited paper). We do not claim that they show phenol formation specifically. We removed the citation from Namysl, since their experiments are performed at higher temperatures. In the revised manuscript, we also note that the oxidation of hydroxybenzyl alcohol by OH produces catechol in very high yield suggesting this maybe an important contributor to the C6 compounds observed in the aerosol.

Line 179: the adduct cannot decompose to phenol +methanol rather phenol + CH2OH radical.

We have changed this to indicate that the radical CH$_2$OH forms first.

Line 180: I do not see a trend in Table 1 and Figure 1.

See updated Table 1. We have run additional experiments and taken a different approach to calibrations.

Line 189: In Table F1 you show oxopentanal not hydroxy oxopentenal. What is correct?
We have made corrections to Table F1.

Line 193: in Figure 4 no hydroxy oxopentenal is formed. As mentioned before, this compound is also not listed in Table F1.

Thank you for pointing out this inconsistency. To avoid further confusion, we have changed all references to this compound to its IUPAC name, 5-hydroxy-4-oxo-2-pentenal, including in Table F1.

Figure 4: oxopropanoic acid is not correct. Oxygen must be on all carbons. Hydroxyacetaldhyde is also not correct, it must be glyoxal.

First point: Oxopropanoic acid + butenedial together have five oxygens. The preceding intermediate compound also has five oxygens. Since this is a decomposition reaction, the number of oxygens retained.

Line 245: HBA rate constant is even lower than that for BA, contrary to your claim that OH-groups increase the rate by factor 4-8. Please comment.

We have revisited the rate constant estimate for HBA. The non-linearity in the measured yield plus the rapid appearance of cresol (major oxidation product) suggest the rate coefficient for the reaction of OH with HBA is quite fast. Consistent with the observations, we find that the rate

coefficient estimated using a group additivity method adequately explains the time dependence of HBA in the chamber.

Table F1: several chemical structures and their names do not agree, e.g. Butenedial, hydroxyoxopropenal, oxopropanoic acid. In some cases also observed m/z and reagent ion are strange. Check carefully

Thank you for your comment. We have made corrections to Table F1.

---

## Author Response (AR2)

**The authors would like to thank the referees for their time reviewing this manuscript. Edits suggested by referee #2 are reflected in the following document in bold.**

Line 198 – 203: in their response to referees the authors refer to Figure 7 in Schwantes et al.. Figure 7 shows that OH-abstraction from the CH2O group in benzaldehyde leads to C6 compounds. Here, the authors claim that OH-addition leads to C6 compounds referring to Schwantes et al. This is not correct. Furthermore, I cannot follow how the mechanism described in this section by the authors should lead to phenol.

**Thank you for your feedback. We have corrected the text to accurately reflect that Schwantes et al. indicates formation of C6 compounds from OH-abstraction from the CH$_2$O group. We have also updated Figure 1 to reflect both potential C6-formation pathways as described in Schwantes et al. and Wang 2015. We hope that this and additional clarifying language helps make this section easier to follow.**

Line 226: upper yield of HBA looks more like 80±9%
**We have updated the yield for the time reported in the text.**

Figure H1: the font size should be increased
**We increased the text size.**

Line 182: delete "as well"
**Deleted.**

---

## Author Response (AR3)

We have reviewed editor's notice of acceptance. We understand that no revisions are needed. We thank the referees for reviewing our manuscript and for the editors for the opportunity to publish in your respected journal.